# Profile of the Nicotinic Cholinergic Receptor Alpha 7 Subunit Gene Expression is Associated with Response to Varenicline Treatment

**DOI:** 10.3390/genes11070746

**Published:** 2020-07-06

**Authors:** Juliana Rocha Santos, Paulo Roberto Xavier Tomaz, Jaqueline Ribeiro Scholz, Patrícia Viviane Gaya, Tânia Ogawa Abe, José Eduardo Krieger, Alexandre Costa Pereira, Paulo Caleb Júnior de Lima Santos

**Affiliations:** 1Laboratory of Genetics and Molecular Cardiology, Instituto do Coracao (InCor), Hospital das Clinicas HCFMUSP, Faculdade de Medicina, Universidade de Sao Paulo, Sao Paulo 05403-904, Brazil; julirocha.farmaceutica@gmail.com (J.R.S.); paulo.biomed@live.com (P.R.X.T.); krieger@incor.usp.br (J.E.K.); acplbmpereira@gmail.com (A.C.P.); 2Smoking Cessation Program Department, Instituto do Coracao (InCor), Hospital das Clinicas HCFMUSP, Faculdade de Medicina, Universidade de Sao Paulo, Sao Paulo 05403-904, Brazil; jaquelineincor@yahoo.com.br (J.R.S.); drapatriciagaya@usp.br (P.V.G.); drataniaogawa@gmail.com (T.O.A.); 3Department of Pharmacology—Escola Paulista de Medicina, Universidade Federal de Sao Paulo, EPM-Unifesp, Sao Paulo 04044-020, Brazil

**Keywords:** varenicline, smoking cessation, gene expression, *CHRNA7*, *CHRNG*

## Abstract

Introduction: Smoking is considered the leading cause of preventable morbidity and mortality worldwide. Studies have sought to identify predictors of response to smoking cessation treatments. The aim of this study was to analyze a possible association of target gene expression for smoking cessation with varenicline. Methods: We included 74 smokers starting treatment with varenicline. Gene expression analysis was performed through the custom RT² Profiler qPCR array assay, including 17 genes. Times for sample collection were before the start of therapy (T0) and two weeks (T2) and four weeks (T4) after the start of treatment. Results: For gene expression analysis, we selected 14 patients who had success and 13 patients resistant to varenicline treatment. Success was considered to be when a patient achieved tobacco abstinence until the fourth week of treatment and resistant was when a patient had not stopped smoking as of the fourth week of treatment. We observed a significant difference for *CHRNA7* gene expression: in the resistant group, samples from T2 and T4 had lower expression compared with T0 (fold change: 0.38, *P* = 0.007; fold change: 0.67, *P* = 0.004; respectively). Conclusion: This exploratory clinical study, searching for a possible predictor of effectiveness for varenicline, reaffirmed the association of the α7 nAChR subunit for nicotine dependence and smoking therapy effectiveness with varenicline.

## 1. Introduction

Smoking is considered the leading cause of preventable morbidity and mortality worldwide. Heart and cerebrovascular diseases, about one fifth of all cancers, lung diseases such as apnea and emphysema and prenatal disorders are associated with tobacco use [1,2,3,4]. 

Nicotine is the main component of addictive tobacco and acts on nicotinic acetylcholine receptors (nAChR), promoting the release of several neurotransmitters and neuroregulators such as dopamine, acetylcholine, epinephrine, norepinephrine, serotonin, β-endorphin, vasopressin and γ -aminobutyric acid (GABA). The main neurotransmitter studied is dopamine. Nicotine increases levels of dopamine in the brain’s mesocorticolimbic region, especially in the circuit known as the “reward system” (ventral tegmental area, nucleus accumbens and prefrontal cortex), generating feelings of pleasure and well-being [5,6]. The process of nicotine addiction is complex and not all mechanisms have been elucidated. In addition to the activation of the reward system, there is a formation of associative memories of hedonic effect with environments and situations related to the use of tobacco, resulting in positive reinforcement related to its consumption [7,8]. 

Varenicline is the newest smoking cessation drug, which has been specifically developed to act on nAChR. It was synthesized based on the chemical structure of cytisine, an alkaloid used in therapy for smoking cessation that is extracted from the Cytisus laburnum L plant [9]. It acts as a partial agonist on nAChRs composed of α4β2 subunits [10], but the drug also acts on other nAChRs, such as nAChR (α3β4), acts weakly on receptors containing α6 subunit and nAChR (α3β2), and exerts full agonist activity on α7 receptors [11]. Figure 1 shows the chemical structure of varenicline.

Interindividual variability in response to drugs for smoking cessation suggests that treatments may be more effective in subgroups of smokers [12,13,14,15]. In the context of personalized medicine, researchers have sought to identify predictors of response to smoking cessation treatments, which are generally involved in pharmacokinetic and/or pharmacodynamic pathways, such as genetic factors, gene expression profiles, proteins and metabolites [16,17,18,19].

In studies of gene expression, obtaining central nervous system samples of patients is not feasible. As an alternative, studies have proposed the analysis of gene expression in peripheral blood [20,21,22]. Although not the target tissue of pharmacodynamics of drugs for smoking cessation, blood has many transcriptional similarities to multiple brain regions [23]. In this context, the main aim of this study was to analyze a possible association of target gene expression in samples from peripheral blood mononuclear cells from patients treated with varenicline.

## 2. Materials and Methods

### 2.1. Patient Sample

A prospective cohort study enrolled 74 patients, men and women, smokers, aged ≥ 18 years, body mass index ≥ 18.5 and <30 kg/m^2^, from the Smoking Assistance Program (PAF), Instituto do Coracao (InCor), Hospital das Clinicas, Faculdade de Medicina, Universidade de Sao Paulo (HCFMUSP), Sao Paulo, SP, Brazil. The protocol was approved by the Ethics Committee for Human Medical Research of the Clinical Hospital of the School of Medicine, University of São Paulo (CAAE 60133816.0.0000.0068; SDC 4341/16/007). Patients were informed of the study and, if they consented, were included and signed an informed consent form in accordance with the Declaration of Helsinki, and all methods were performed according to good clinical practice guidelines.

Exclusion criteria were: patients with hepatic, renal and gastrointestinal disorders that compromised drug metabolism and elimination; patients who had taken cytochrome P450 enzyme-inducing or inhibiting drugs in the previous 6 weeks; alcoholic patients and drug users; those with unstable psychiatric illnesses; women at risk of pregnancy; patients with contraindications to the treatment with varenicline mentioned above. 

The patients were followed by the InCor-PAF team, and pharmacological treatment with varenicline was conducted for 12 weeks in dosages according to the package insert (from the 1st to the 3rd day, 0.5 mg once a day; from the 4th to the 7th day, 0.5 mg twice daily; and from the eighth day until the end of treatment, 1 mg twice daily). Patient data were collected in four visits at the following times: T1 (initial visit), T0 (before the start of pharmacological treatment), T2 and T4 (two and four weeks after the start of pharmacological treatment, respectively).

The following treatment outcomes were considered: success, when the patient achieved tobacco abstinence until T4, and resistant, when the patient could not stop smoking by the end of T4. Smoking abstinence was confirmed by the measurement of exhaled carbon monoxide in all visits. Patients with carbon monoxide values <4 ppm were considered abstinent.

The customized assay for the analysis of gene expression was a major financial investment in this research and, therefore, we used it to analyze only the samples of patients treated with varenicline, since it is the newest, most effective and most costly drug in the anti-smoking treatment. The 17 genes with the greatest potential for association with response to treatment with varenicline were chosen, according to previous studies [16,18,24,25].

Patients who started treatment with varenicline underwent blood collection for RNA in three time periods: T0, T2 and T4. For blood collection, PAXgene Blood RNA tubes (PreAnalytiX GmbH, Feldbachstrasse, Hombrechtikon, Switzerland) were used, and, after, they were incubated for 2 h at room temperature to ensure complete lysis of blood cells and then stored at −20°C until RNA was extracted from the samples.

As for the sample number for the gene expression assay, of the 74 patients treated with varenicline, samples were chosen from 27 patients (14 in the success group and 13 in the resistant group), a number based on previous studies [26,27], who correctly followed fasting, took the medication at the agreed times and attended all collection times. Altogether, there were 81 samples (from the 27 patients) due to the 3 treatment times (T0, T2 and T4). However, some samples did not show good results for the analysis of gene expression and we performed the expression test for 60 samples of different times. Eight samples did not show amplification results, and we ended up with 52 samples in all (9 resistant from time T0, 7 resistant from T2, 8 resistant from T4, 9 successes from T0, 8 successes from T2 and 11 successes from T4) from 14 patients in the success group and 13 in the resistant group. This final sample size for each analysis group is comparable to previous studies [27].

### 2.2. RNA Extraction, cDNA Synthesis and gene Expression Assay

The RNA purification and concentration process was performed using the PAXgene Blood RNA Kit (PreAnalytiX GmbH, Feldbachstrasse, Hombrechtikon, Switzerland). The cDNA synthesis process was performed using the RT^2^ HT First Strand Kit (Qiagen, Frederick, Maryland, USA) as described by the manufacturer. All cDNA samples were analyzed for quality by quality-control assay (QC disc, Qiagen).

The expression protocol was performed by real-time PCR using the Rotor Gene 6000 (Corbett) equipment with a custom RT² Profiler PCR Array assay with RT² FAST SYBR Green/ROX qPCR Master Mix (Qiagen, Frederick, Maryland, USA). For gene expression assay, 4 housekeeping genes (*B2M*, *GAPDH*, *HPRT1* and *ACTB,*
Appendix A) were used and 17 target genes were analyzed (*CHRNA3*, *CHRNA4*, *CHRNA5*, *CHRNA6*, *CHRNA7*, *CHRNB2*, *CHRNB3, CHRNB4*, *CHRNG*, *DRD1*, *DRD2*, *DRD3*, *DRD4*, *HTR3A*, *HTR3B*, *COMT*, *SLC6A3;*
Appendix A). Real-time PCR was programmed for PCR cycling under the following conditions: 1 cycle at 95 °C lasting 10 min; 40 cycles at 95 °C for 15 s and 60 °C for 30 s. Using the real-time cycler software (Qiagen Frederick, Maryland, USA), the threshold cycle was adjusted and the CT data were spread out for statistical analysis.

### 2.3. Statistical Analysis

Regarding demographic characteristics, continuous variables were presented as mean and standard deviation and categorical variables as frequencies. For the analysis of expression data, the 2^−∆∆CT^ method was used [28]. Values of CT were normalized by housekeeping gene expression using the following formula: ∆CT = (CT_gene target_ - the mean of CT_housekeeping genes_); ∆CT values are shown in Appendix A. Shapiro–Wilk and Kolmogorov–Smirnov tests were performed to verify the normality of gene expression levels (2^−∆CT^ values). The 2^−∆CT^ median values of each group were used for Mann–Whitney hypothesis testing (because the distribution of values is not normal). The values of ∆∆CT (median of ∆CT_test group_ - median of ∆CT_reference group_) were obtained for the calculation of fold change. The fold change values were generated using the following formula: 2^−∆∆CT^. The heat map graph was constructed using Gene-globe software (Qiagen, Frederick, Maryland, USA) to verify the differences in the expression profile of the studied genes according to the times (T0, T2 and T4) and according to the effectiveness of varenicline treatment. All statistical analysis was performed using SPSS software (version 20, IBM corp, Armonk, NewYork, USA), considering a significance level of *P* < 0.05.

## 3. Results

From the 74 patients treated with varenicline, we selected samples from 14 patients from the success group and 13 from the resistant group. Table 1 shows the clinical and demographic characteristics of the patients selected for the gene expression assay. 

Table 2 shows fold change and the association analysis between time periods and outcome groups. Of the 17 genes chosen to test treatment response, four showed amplification values which were more acceptable for the quality of the interpretation, more specifically CT values <33. We observed significant differences for *CHRNA7* gene expression: in the resistant group, samples from T2 and T4 had lower expression compared with samples from T0 (fold change: 0.38, *P* = 0.007; fold change: 0.67, *P* = 0.004; respectively). For *CHRNG* gene expression, samples from T2 of the success group had lower expression compared with samples from T2 of the resistant group (fold change = 0.77, *P* = 0.006). We performed statistical correction for age, cigarettes per day and gender. These variables do not impact the identified significant data.

Appendix A shows a heat map of gene expression magnitude. The heat map shows that there was no differentiation in gene expression profiles in a joint gene analysis between outcome and treatment times.

## 4. Discussion

The present clinical and exploratory study shows the results of *CHRNA7* and *CHRNG* gene expression according to the collection times and the treatment outcome. Samples of the resistant group at times T2 and T4, that is, with varenicline, show lower levels of *CHRNA7* gene expression compared to T0 samples (before starting pharmacological treatment). The *CHRNA7* gene encodes the nAChR alpha7 subunit. These results suggest that the downregulation of α7 may have been responsible for the inability to stop smoking in individuals of the resistant group, which was not observed in samples of the success group.

The α7 subunit has great importance, is present in several tissues and has had a consistent association in several studies with the process of nicotine dependence (ND) cognitive and immunological processes [29]. Studies have also shown that there is a locus containing a truncated *CHRNA7* gene duplication, composed of 5–10 *CHRNA7* gene exons, preceded by four unique exons and a two-base-pair deletion in exon 6, which they named *CHRNFAN7*. Most individuals have one or two copies of this gene and it is rare for this copy to be absent [30].

In the brain of Sprague Dawley rats, Ryan and Loiacono [31] detected transcribed α7 levels at various locations, such as the substantia nigra pars compacta (SNpc), substantia nigra pars reticular (SNpr), ventral tegmental area (VTA), cortex and hippocampus. The rats submitted to chronic nicotine treatment showed higher levels of α7 subunit mRNA in SNpc, SNpr and VTA compared to the control group. VTA is part of the mesolimbic reward system, which is widely studied in ND processes [5,6]. In the present study (in peripheral blood mononuclear cell—PBMC samples), the resistant patient group also showed higher α7 expression when only nicotine was present (T0). By introducing varenicline, expressions were decreased at the T2 and T4 times, probably due to the competitive antagonism of varenicline with nicotine at the receptor.

Benhammou et al. [32] conducted a study with post-mortem brain samples from smokers and blood samples from smokers and non-smoker volunteers. Through the use of radiolabelled nicotine in brain samples, they showed that nicotine binding sites increased in proportion to the number of cigarettes per day. The same study analyzed the gene expression and protein in lymphocytes and polymorphonuclear cells for a subset of nicotinic receptor subunits; these cells, as well as brain tissue, exhibited higher numbers of high-affinity nicotine binding sites in smokers compared with non-smokers.

The varenicline in functional studies showed full agonist activity at the α7 receptors [11]. In one study, chronic treatment was performed with varenicline and/or nicotine in mice (10 days). Brain and plasma drug levels, and tolerance and expression of four nAChR subtypes after an acute dose of nicotine, were checked using autoradiography. Upregulation of α4β2 receptors, due to chronic varenicline treatment, was similar to that promoted by nicotine. Both varenicline and nicotine promoted downregulation of α6β2 nAChR. Varenicline significantly increased α3β4 and α7 levels, while nicotine had a minor effect on these sites. The combination of nicotine and varenicline was similar to varenicline alone for the α3β4 sites, while for the α7 site the combination promotes upregulation in fewer brain regions compared to varenicline monotherapy [33]. In the present study, resistant group patients showed similar effects associated with the combined use of varenicline and nicotine (due to smoking). *CHRNA7* expression levels were not maintained as they were in patients who were successful in treatment. However, the success group was only using varenicline in the absence of nicotine between times T2 and T4, which suggests that this downregulation may occur in the response subgroup.

Several studies showed differences in α7 gene expression in sample patients with clinical psychiatric diagnosis compared to healthy individuals. Kunii et al. [34] showed a difference in α7 expression in dorsolateral prefrontal cortex postmortem tests in patients with depression, anxiety and bipolar disorder, and several studies showed lower α7 expression levels in patients with schizophrenia [27,35,36]. Studies of individuals with schizophrenia are great models for understanding the importance of the α7 receptor in cognitive processes in ND and how these mechanisms are closely related. Several studies showed that there is a higher prevalence of smoking in individuals with schizophrenia compared to the general population [37]. Mexal et al. [27] performed a study with post-mortem brain samples from four distinct groups: control, smoker control, non-smoker schizophrenia patients and smoker schizophrenia patients. The study performed mRNA and protein expression assays by real-time quantitative PCR and western blot, respectively. The PCR probes were designed to amplify the complete *CHRNA7* gene and its duplicate copy, the *CHRFAM7A* gene. Non-smokers with schizophrenia present reduced mRNA and protein expression compared to smoker schizophrenic patients. Non-smokers with schizophrenia present lower expression than non-smoker controls, but this difference was statistically significant only at the mRNA level. Other brain post-mortem studies showed lower nicotinic receptor expression in the cortex and hippocampus of individuals with schizophrenia compared to healthy controls [35,36,38,39]. *CHRFAM7A*, like *CHRNA7*, is polymorphic, and variants in this gene were associated with auditory pathology in cases of schizophrenia. Differentiated processes in transcription mechanisms may contribute to abnormal α7 functioning, such as promoter variations, alternative splicing and/or linkage disequilibrium with SNPs (single-nucleotide polymorphisms) located at the neuroregulin locus (NRG1) on chr8 (the gene encoding the cell adhesion molecule involved in human synaptic neuroplasticity). These processes may be involved in schizophrenia and bipolar disorder endophenotypes [30,40,41,42,43,44,45]. Nicotine seems to improve hearing in individuals with α7-associated hearing deficit in schizophrenia, allowing better filtering of external auditory stimuli. Nicotine improves cognition in schizophrenia, and alternative agents that activate the nicotinic receptor have been tested and these compounds improved attention, working memory, and negative symptoms in a complementary study of non-smoking patients with schizophrenia [46].

The studies cited show us strong evidence of the association of α7 with ND, cognitive processes and varenicline response. These findings may explain our results regarding decreased expression of α7 in the group resistant to treatment. The success group patients maintained gene expression levels in response to varenicline, while, in the resistant group, there was a downregulation. We suggest that, possibly, this decrease may have been responsible, at least in part, for patients failing to quit smoking. Nicotine may be compensating for an α7 deficit that resistant individuals may have. However, we do not know what the α7 expression would be in these individuals in the absence of smoking. Another possibility is that the association occurred by chance: the response to varenicline in the resistant group could have resulted in the downregulation of *CHRNA7*, due to the genetic or epigenetic polymorphism at *CHRNA7* or a regulator of *CHRNA7*, but without influencing lapse/relapse likelihood.

For the *CHNRG* gene, there are a paucity of studies in the literature. This encodes the γ subunit of nicotinic receptors. Most studies have indicated an association of the subunit with skeletal muscle disorders [47,48,49,50]. However, King et al. [16] found the association of polymorphisms in this gene with nausea in varenicline treatment. Keskitalo-Vuokko et al. [51] found the association of SNPs in the *CHRNG-CHRND* gene cluster loci (adjacent genes, the last encoding the nAChR delta subunit) with cotinine level. Saccone et al. [52] found the association of four loci in the *CHRND-CHRNG* cluster with ND. In another study, variants in *CHRND-CHRNG* genes showed a modest association with the risk of ND in African American samples [53]. In a study in which a rat model of schizophrenia was used, in the clozapine-treated animal group *Chrng* downregulation was found in the nucleus accumbens. In this same study, Santoro et al. failed to formulate a hypothesis for the finding due to the lack of information in the literature focusing on ND or cognitive performance [54].

There are limitations to the study. First, the gene expression assay was performed with peripheral blood samples rather than using central nervous system (CNS) samples, which could show us a clearer and more direct relationship of the drug to the response pathway in the brain. However, obtaining brain biopsy samples is not feasible. Cerebrospinal fluid could be used, but it is a painful procedure for the patient. In contrast, studies have already shown that there is similarity between the gene expressions of CNS cells and peripheral blood cells for some of these nicotinic receptor genes [27]. Second, the sample size of the success and resistant patient groups is small, but this patient cohort is well phenotyped for antismoking therapy. Third, no gene expression assay was performed for samples from non-smokers, which could provide more parameters for comparison with resistant and success groups. Fourth, we did not achieve the required quality (measurable amplification) in potentially important gene analysis.

## 5. Conclusions

In this exploratory clinical study searching for a possible predictor of the effectiveness of varenicline, we reaffirmed the association of the α7 nAChR subunit with nicotine dependence and treatment effect with varenicline. This data could contribute to the development of personalized therapy with varenicline. However, additional studies are needed to further understand the role of the α7 and γ nAChR subunits in smoking cessation therapy using varenicline.

## Figures and Tables

**Figure 1 genes-11-00746-f001:**
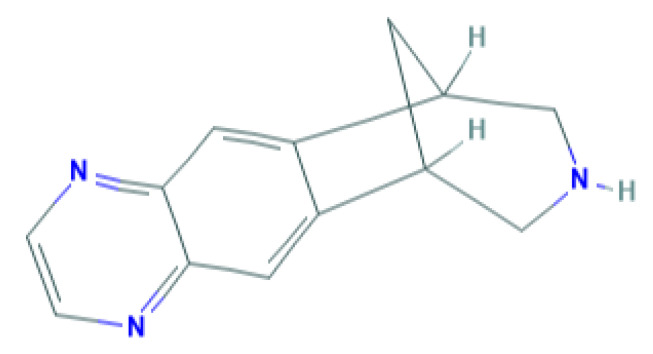
Chemical structure of varenicline. Source: PubChem database.

**Table 1 genes-11-00746-t001:** Clinical and demographic characteristics of patients treated with varenicline according to the treatment outcome.

	Resistant (*n* = 13)	Success (*n* = 14)	*p* Value
Age (years)	55 ± 11	45 ± 12	0.045
Gender, female (%)	53.8	35.7	0.45
Race, white (%)	76.9	78.6	1.00
Body mass index (Kg/m2)	27.1 ± 4.6	27.3 ± 5.4	0.92
Cigarettes/day	20 ± 6	15 ± 6	0.06
FTND	6.3 ± 2.2	5.9 ± 1.9	0.58
FTND, ≥6 (%)	53.8	64.3	0.70
Hypertension (%)	23.1	7.1	0.33
Dyslipidemia (%)	23.1	28.6	1.00
Diabetes mellitus type 2 (%)	23.1	0.0	0.10
Depression (%)	7.7	7.1	1.00
Anxiety (%)	30.8	35.7	1.00

FTND = Fagerström test for nicotine dependence (0 to 10 point scale).

**Table 2 genes-11-00746-t002:** Fold change and association analysis between time periods and outcome groups.

	Resistant T2 *vs.* T0		Resistant T4 *vs.* T0		Resistant T4 *vs.* T2	
Genes	2^−∆∆CT^ (95% CI)	*p* Value	2^−∆∆CT^ (95% CI)	*p* Value	2^−∆∆CT^ (95% CI)	*p* Value
***CHRNA5***	0.83 (0.34–2.33)	0.57	1.42 (0.30–2.69)	0.95	1.72 (0.34–3.03)	0.89
***CHRNA7***	0.38 (0.20–0.93)	**0.007**	0.67 (0.31–1.01)	**0.004**	1.78 (0.58–2.87)	0.37
***CHRNG***	0.83 (0.54–1.92)	0.37	0.74 (0.43–1.90)	0.44	0.90 (0.50–1.58)	0.36
***COMT***	0.91 (0.65–1.49)	0.92	0.87 (0.56–1.40)	0.63	0.95 (0.57–1.42)	0.73
	**Success T2 *vs.* T0**		**Success T4 *vs.* T0**		**Success T4 *vs*. T2**	
**Genes**	2^−∆∆CT^ **(95% CI)**	***p* Value**	2^−∆∆CT^ **(95% CI)**	***p* Value**	2^−∆∆CT^ **(95% CI)**	***p* Value**
***CHRNA5***	2.38 (0.33–6.23)	0.37	0.97 (0.37–3.10)	0.61	0.41 (0.20–2.68)	0.69
***CHRNA7***	1.16 (0.31–4.06)	0.64	0.92 (0.23–3.16)	0.91	0.79 (0.29–2.04)	0.55
***CHRNG***	0.99 (0.45–2.27)	0.49	1.17 (0.56–2.81)	0.62	1.18 (0.72–2.11)	0.33
***COMT***	1.23 (0.55–1.72)	0.92	1.05 (0.55–1.55)	0.68	0.85 (0.56–1.59)	0.41
	**Success *vs.* resistant T0**		**Success *vs.* resistant T2**		**Success *vs.* resistant T4**	
**Genes**	2^−∆∆CT^ **(95% CI)**	***p* Value**	2^−∆∆CT^ **(95% CI)**	***p* Value**	2^−∆∆CT^ **(95% CI)**	***p* Value**
***CHRNA5***	0.91 (0.28–2.50)	0.75	2.60 (0.36–5.06)	0.47	0.62 (0.34–2.85)	0.85
***CHRNA7***	0.64 (0.21–1.82)	0.13	1.97 (0.61–4.23)	0.09	0.88 (0.42–2.14)	0.74
***CHRNG***	0.65 (0.26–1.69)	0.12	0.77 (0.40–1.09)	**0.006**	1.03 (0.50–1.71)	0.79
***COMT***	0.86 (0.61–1.64)	0.90	1.16 (0.61–1.62)	0.49	1.04 (0.64–1.69)	0.93

T0 = initial time before pharmacological treatment, T2 = two weeks after pharmacological treatment, T4 = four weeks after pharmacological treatment. *p* Value (association analysis between the 2^−∆CT^ medians of the groups). 95% CI = 95% confidence interval.

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
