# Peer review of "Profile of the Nicotinic Cholinergic Receptor Alpha 7 Subunit Gene Expression is Associated with Response to Varenicline Treatment"

_genes, 2020, doi:10.3390/genes11070746_

Round 1
Reviewer 1 Report
Since previous work by your group has found that the sequence of α4β2 rs1044396 is a target for varenicline induced ND and smoking cessation success, I am curious why you didn't include their mRNA in this study of the peripheral blood cells' AChR mRNA expression. It clearly would require a lot more work, but you did include the α5 subunit for what reason I could not figure. In your current study, the emphasis is on expression of α7 and CHRNG gamma mRNA (what are they doing in lymphocytes?) and how that expression is altered by varenicline treatment over 4 weeks. The affinity of varenicline is greater for the α4β2 receptor compared to α7, but a direct comparison of how with the qPCR technology the two stack up would be useful, since you would like us to believe that both are players in whether a patient will be positively affected by varenicline treatment. As with much of central nervous system mechanisms, nothing is straight forward or linear, but I think the qPCR of the different AChR mRNAs in the periphery you have introduced is a good starting way to go. I am impressed that the effect of varenicline is reflected on the those receptors in the lymphocytes. If only the technology would come along to allow you to get some specific cellular material from the various brain regions associated with addiction. You work on a tough problem. You should show the readers the structure of varenicline! Looks just like acetylcholine, no? Keep it up!
Author Response
Dear reviewer,
We thank you very much and appreciate your comments and suggestion. Point-to-point responses to your comments are also highlighted in blue in this letter (attached).
Paulo C J L Santos, PhD. Department of Pharmacology, Escola Paulista de Medicina, Universidade Federal de Sao Paulo Unifesp, São Paulo, SP, Brazil.

Reviewer 2 Report
Authors please see attached file.
This reviewer thinks that 3 of the 4 limitations discussed in this manuscript should be addressed before publication, which is unfortunate because this is important work and the manuscript is well written.

Author Response

(The authors gave the same response as above.)

Round 2
Reviewer 2 Report
This reviewer is satisfied with the author's revisions.